# Changes in Forest Stand and Stability of Uropodine Mites Communities (Acari: Parasitiformes) in Jakubowo Nature Reserve in the Light of Long-Term Research

**Jerzy Błoszyk** [1,2], **Agnieszka Napierała** [1,*] ⬤, **Marta Kulczak** [1] and **Michał Zacharyasiewicz** [1]

1. Department of General Zoology, Faculty of Biology, Adam Mickiewicz University in Poznań, Uniwersytetu Poznańskiego 6, 61-614 Poznań, Poland; bloszyk@amu.edu.pl (J.B.); markul12@st.amu.edu.pl (M.K.); zacharyasiewicz@amu.edu.pl (M.Z.)
2. Natural History Collections, Faculty of Biology, Adam Mickiewicz University in Poznań, Uniwersytetu Poznańskiego 6, 61-614 Poznań, Poland
* Correspondence: agan@amu.edu.pl

**Abstract:** The current study has been conducted for over 40 years (between 1981 and 2022) in a natural forest reserve in Jakubowo (western Poland). The material for the analysis was collected in three permanent monitoring ground plots with different vegetation cover, humidity and degree of shade. The major aim of the study was to analyze the changes in the species composition and abundance in uropodine (Acari: Parasitiformes) mite communities that occurred in the three ground plots in Jakubowo over 40 years. The second goal was to assess the stability of the species composition and the number of Uropodina mites in the examined communities. The most important phenomenon observed during the research period was a considerable decrease in the abundance of Uropodina in ground litter and soil, and the second was the loss of stenotopic and rare species. Similar observations are also presented in previous studies, which embraced 36 years of research period (between 1978 and 2013). Now, this unfavorable trend, which is caused by anthropogenic disturbances in the environment, has become permanent.

**Keywords:** anthropogenic disturbances; environmental monitoring; forest reserve; long-term research; natural succession; oak-hornbeam forests; stability of mite communities; Uropodina

## 1. Introduction

Many years of acarological observations in one location focusing only on Uropodina (Acari: Parasitiformes) mites, which have been carried out for over 40 years in Jakubowo (oak-hornbeam nature reserve) in western Poland, are undoubtedly unique in this kind of research on a global scale. The research in this nature reserve had been conducted since April 1979 on three permanent monitoring plots with different vegetation cover, humidity and degree of shade [1–4]. From the beginning until now, the supervision over the course of these observations was carried out by the initiator of the research (the first author). The research was and still is carried out all the time with the same methods, and the extraction of the fauna takes place with the same devices. At the same time, photographic documentation of the examined plots is prepared, which allows for a more precise assessment of the changes that have occurred in a given plot and then to draw conclusions about the impact of these changes on the species composition and the number of mites from the suborder Uropodina. In addition to mites, in Jakubowo, changes in communities of terrestrial snails were also monitored [5]. However, most publications concerning Jakubowo focus on mites from the suborder Uropodina [1–4,6,7]. The current article is a continuation of previous research because it takes into account the subsequent years of research against further changes in the vegetation cover on the examined plots.

Over the course of 43 years, the research plots have undergone changes resulting from both the natural succession of plant cover and changes in the management of dead wood

in nature reserves in Poland. Until the end of the 1970s, dead wood was removed from nature reserves. Thus, the mites were deprived of one of the most important microenvironments, the presence of which determines the high biodiversity of this group of mites in forests [2,8,9]. Leaving dead trees in the reserve considerably changed the conditions for the functioning of the entire ecosystem, significantly expanding the range of niches for all fauna, including Uropodina. The increase in the amount of dead wood in the area of this reserve has been observed especially in recent years (since 2000), and it seems to be one of the results of the natural reconstruction of the forest stand caused mainly by felling of old monumental beech trees due to windbreaks. These changes for each monitoring area are described in more detail later in this article.

The major aim of the research presented in the current study was to analyze the changes in the species composition and abundance of uropodine mite communities that occurred in three ground plots in Jakubowo over 40 years. The second goal was to assess the stability of the species composition and the abundance of Uropodina communities in the analyzed period in the selected areas of the reserve.

## 2. Materials and Methods

The samples from Jakubowo nature reserve were collected in an oak-hornbeam forest, in three selected monitoring ground plots (each of them covered an area of 625 m$^2$). In each research period, the mites were collected with the same methods: they were mainly litter and soil samples collected with a biocenometer from a depth of 10 cm. In each of the selected plots, a series of 10 samples were collected once at two-week intervals (between April and November in the period 1981–1982) or monthly in subsequent years of the study (i.e., 2005, 2006, 2012, 2014, 2016 and 2022). In 2012 and 2014, no samples were collected from plot J-II, focusing solely on the remaining plots. The mesofauna was extracted with Tullgren funnels for 3 or 5 days, depending on the degree of humidity of the samples, and the specimens were then preserved in 75% ethyl alcohol. The identification of the Uropodina is based on the original description keys [10–13]. In each phase of the research project, all specimens were verified by the first author. The specimens were deposited in the Natural History Collections (Faculty of Biology at Adam Mickiewicz University in Poznań).

The study rests upon the metadata stored in the Soil Fauna Bank in AnalizaTor 2.0 software by Desmodus (Poznań, Poland) (Natural History Collections). The full list of mite species found there is given in the publication [7].

### 2.1. Data Analysis

Due to the fact that the number of collected samples is different for each period, the dynamics of the changes in the analyzed mite communities is illustrated with a scale of dominance (D) and frequency (F) of occurrence [1,14]. The scale has the following classes: dominance D5 eudominants (>30%), D4 dominants (15.1%–30.0%), D3 subdominants (7.1%–15.0%), D2 recedents (3.0%–7.0%) and D1 subrecedents (<3%); frequency F5 euconstants (>50%), F4 constants (30.1%–50%), F3 subconstants (15.1%–30.0%), F2 accessory species (5.0%–15.0%) and F1 accidents (<5%) [1]. Statistically significant differences between the abundance of all Uropodina in two research periods were established with the Mann–Whitney U-test, and statistically significant differences between the abundance of four most frequent species in the examined plots in each research period were established with the (ANOVA) Kruskal–Wallis rank test.

The community similarity of the species composition for Uropodina mites inhabiting each ground plot was calculated by means of the Marczewski–Steinhaus species similarity index: $S = c/(a + b - c)$, where $c$ is the number of species present in both compared communities, and $a$ and $b$ stand for the total number of species in each community [14]. The full joining analysis, which uses the most distant neighbors, was used to draw the dendrogram. The analyses were calculated with AnalizaTor 2.0 software (Poznań, Poland).

## 2.2. Changes in Plant Cover in Monitored Ground Plots

Jakubowo nature reserve, which covers an area of 4.22 ha, is located in western Wielkopolska (Greater Poland) (52°48′17″ N 16°28′67″ E) (Figure 1). A more detailed description of the reserve is presented in earlier studies [1,4]. On the examined ground plots, natural reconstruction of the stand was observed as a result of the loss of the oldest beech trees caused by windbreaks and the development of the undergrowth.

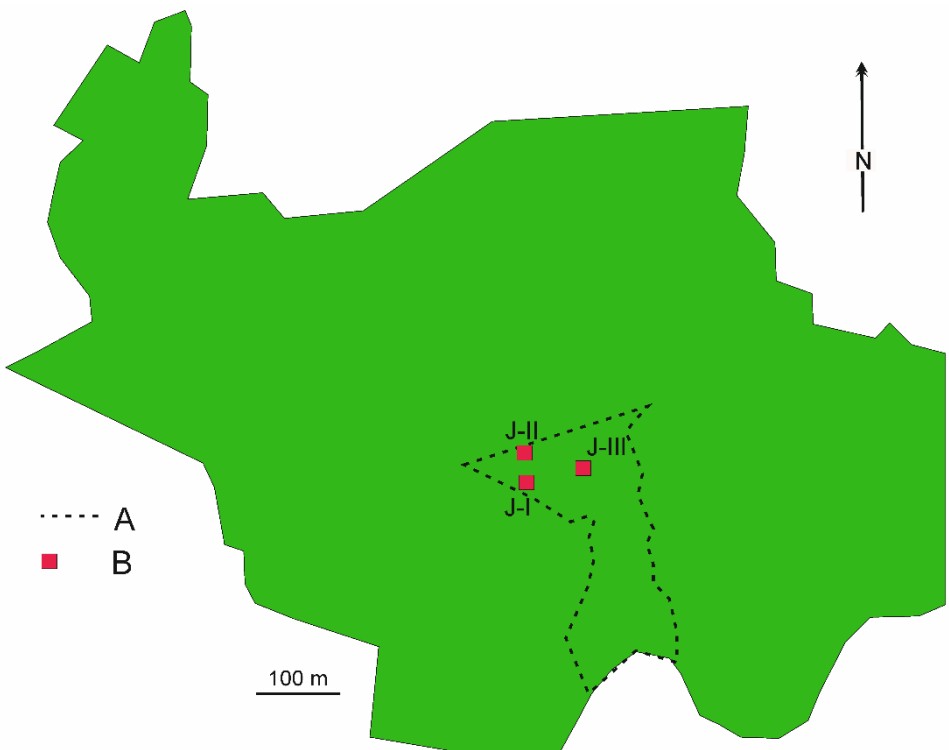

**Figure 1.** Exact location of examined ground plots J-I, J-II, J-II (red square) in Jakubowo nature reserve against whole forest area of the reserve.

The photographs presented below show the changes in the vegetation in the examined ground plots in the period between April 1979 and February 2022, i.e., over 43 years. The characteristics of each surface describe the extent of these changes. The areas differed not only in the type of the plant cover but also in the dynamics of stand succession over the years of research.

Plot J-I (52°48′28″ N 16°28′52″ E) (Figure 1) is a typical oak-hornbeam forest area (*Galio-sylvatici-Carpinetumstachyetosum* var. with *Fagus sylvatica*). In the first period of research (1981–1982), this area was adjacent to a large cut-over area. This plot was the driest area, with considerable daily and seasonal temperature fluctuations. The stand consisted mainly of old monumental beech (*Fagus sylvatica* L.), with a slight mixture of hornbeam (*Carpinus* L.) and oaks (*Quercus* L.). There was no undergrowth (Figure 2A). The ground flora comprised mainly grass (95%) with a slight mixture of sedge (Figure 3A). The litter contained mainly old beech leaves (75%) and oak, hornbeam and sedge leaves.



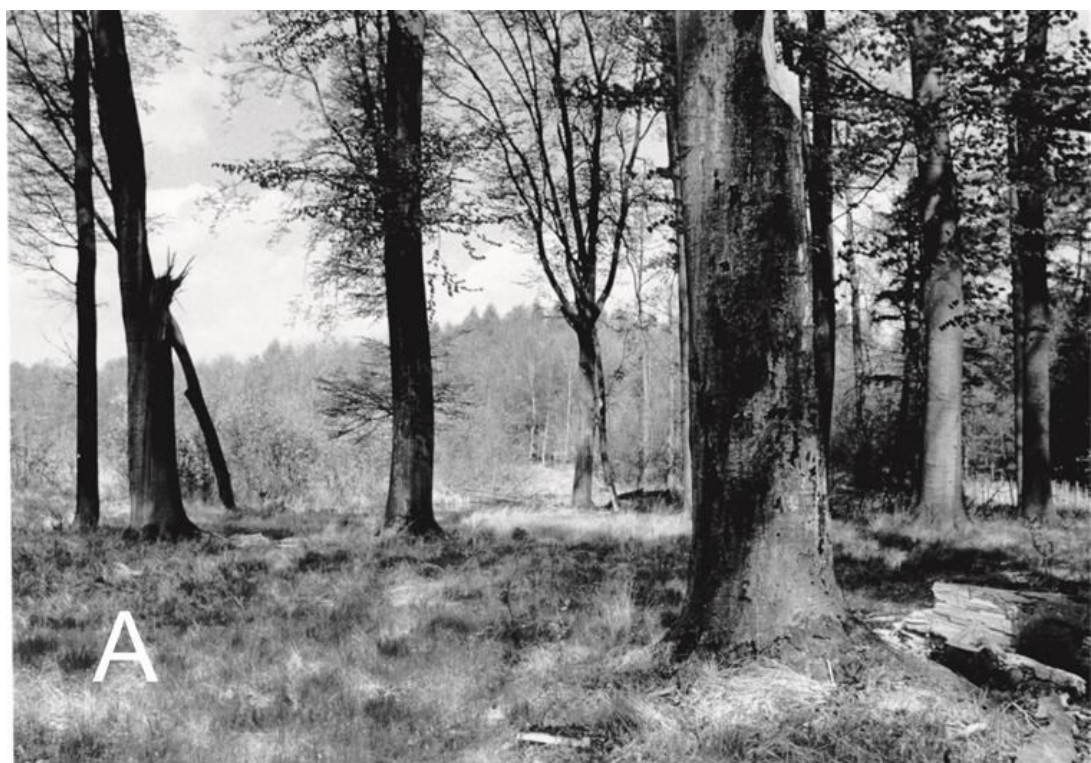

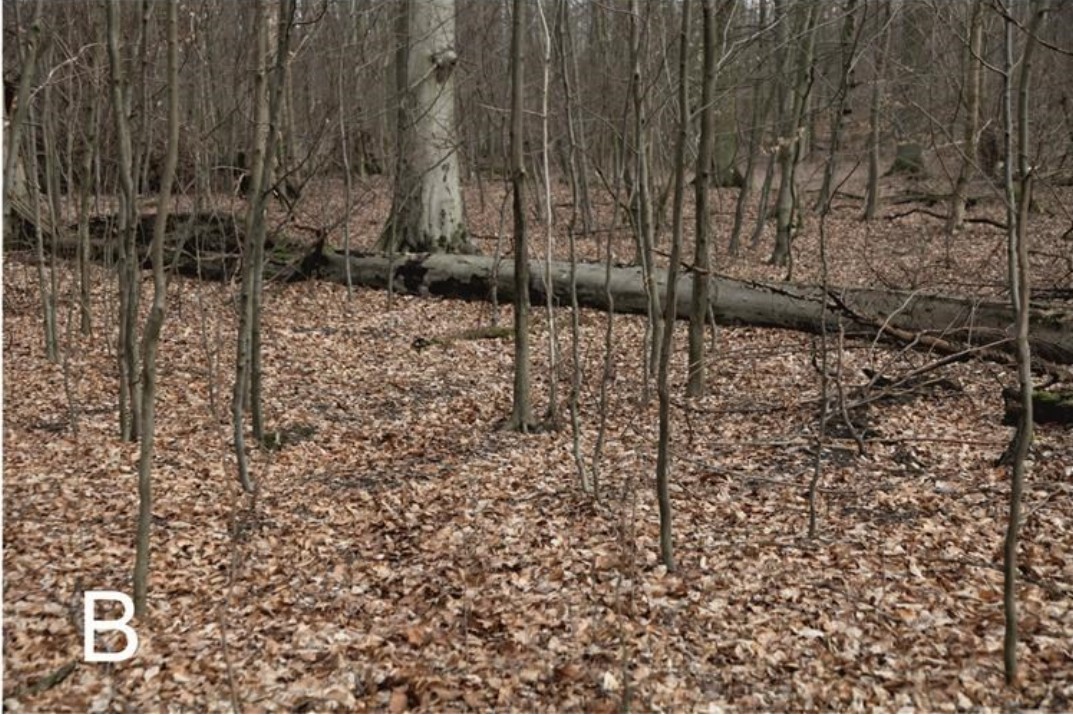

**Figure 2.** Changes in ground flora in plot J-I: (**A**)—In 1979, (**B**)—Present (2022).

After 1985, in the cut-over area, new beech trees began to grow, which are now over 40 years old. Due to windbreaks, in this ground plot, the old beeches started to collapse one by one. In the resulting gaps, a gradual renewal of beech trees can be observed, which currently form a fairly dense undergrowth (Figure 2B). This, in turn, causes an increase in shade and therefore increases the soil moisture. The increase in shade has eliminated the grassy undergrowth. Currently, the litter consists mainly of a thick layer of fallen beech leaves (Figure 3B).

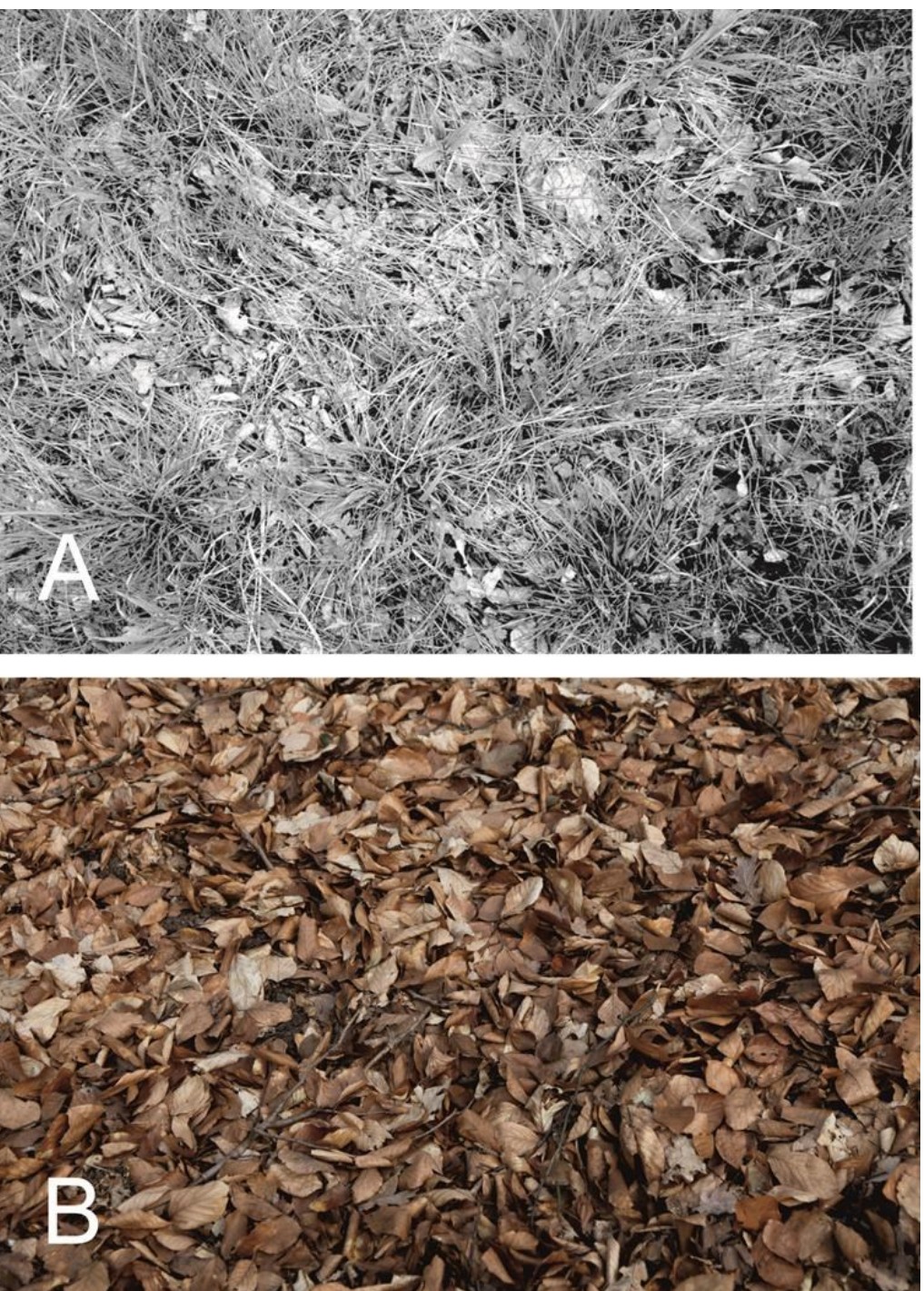

**Figure 3.** Changes in ground cover in plot J-I: (**A**)—Grass and sedge (1979); (**B**)—Now only a thick layer of fallen beech leaves (2022).

Plot J-II (52°48′28″ N 16°28′56″ E) (Figure 1) is an area mainly with oak-hornbeam forest with beech trees (*Galio-sylvatici-Carpinetum stachyetosum* var. with *Fagus sylvatica/Caricetum acutiformis*). It is located at a shallow terrain hollow, where water used to accumulate during rainy periods. The last surface flooding in this area was observed in 1982. There are also hazel bushes (*Corylus* L.), and earlier, there was quite dense common dogwood (*Cornus sanguinea* L.) and not too much grass. At present, however, in this area, there is almost no ground flora (Figure 4A).

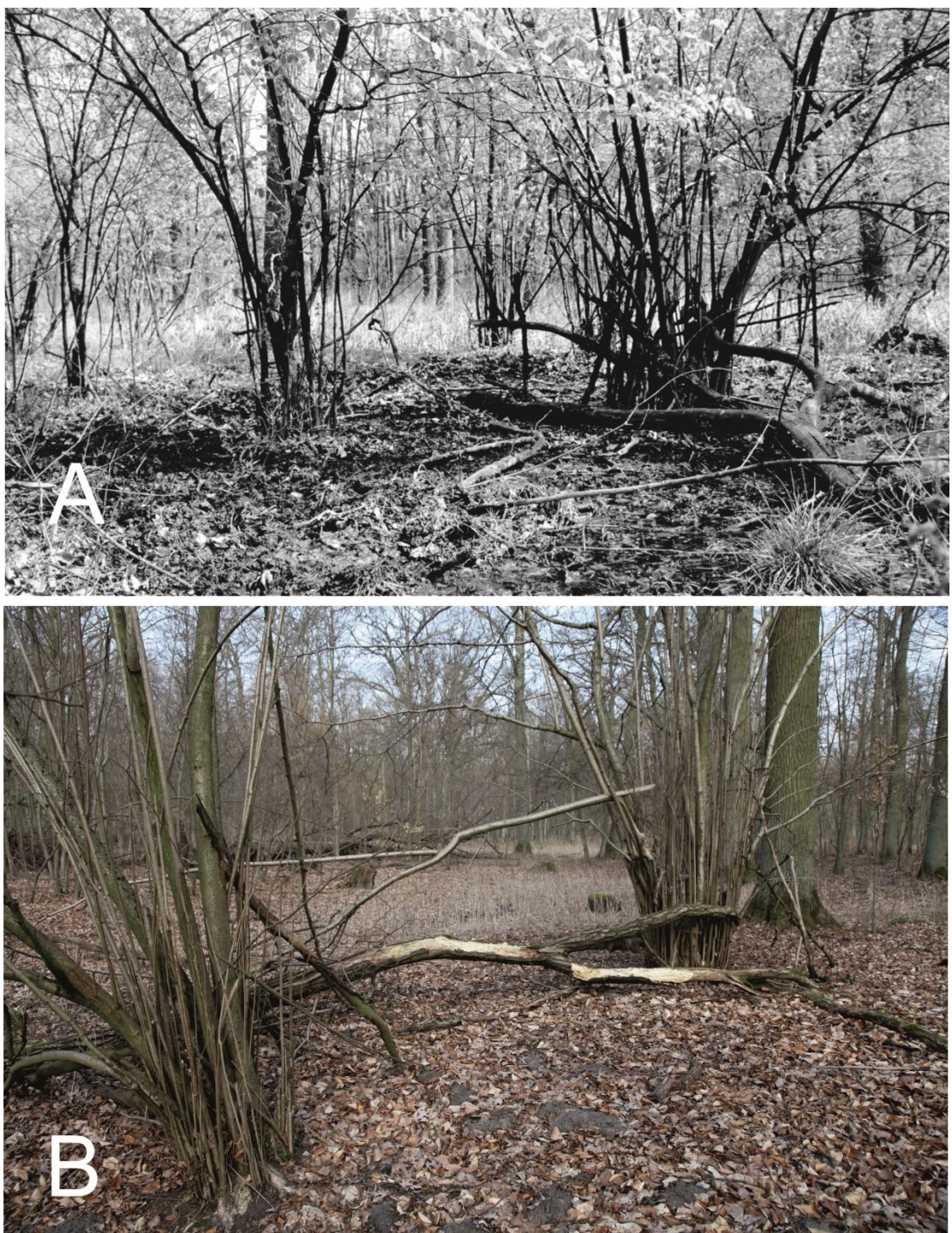

**Figure 4.** Changes in ground flora in plot J-II: (**A**)—In 1979, (**B**)—Present (2022).

In this area, only the bushes of the hazel (*Corylus avellana* L.) are still present, whereas the common dogwood has been lost completely. Additionally, the sporadic tufts of the sedge (*Carex*) are not present now (Figure 4B).

Plot J-III (52°48′29″ N 16°28′57″ E) (Figure 1) is a moderately humid area of an oak-hornbeam forest with beech trees (*Galio-sylvatici-Carpinetum stachyetosum* var. with *Fagus sylvatica*) (Figure 5A). The main changes that have taken place in this area are the higher percentage of dead wood matter and the presence of new dense undergrowth, consisting mainly of young beech trees (Figure 5B).

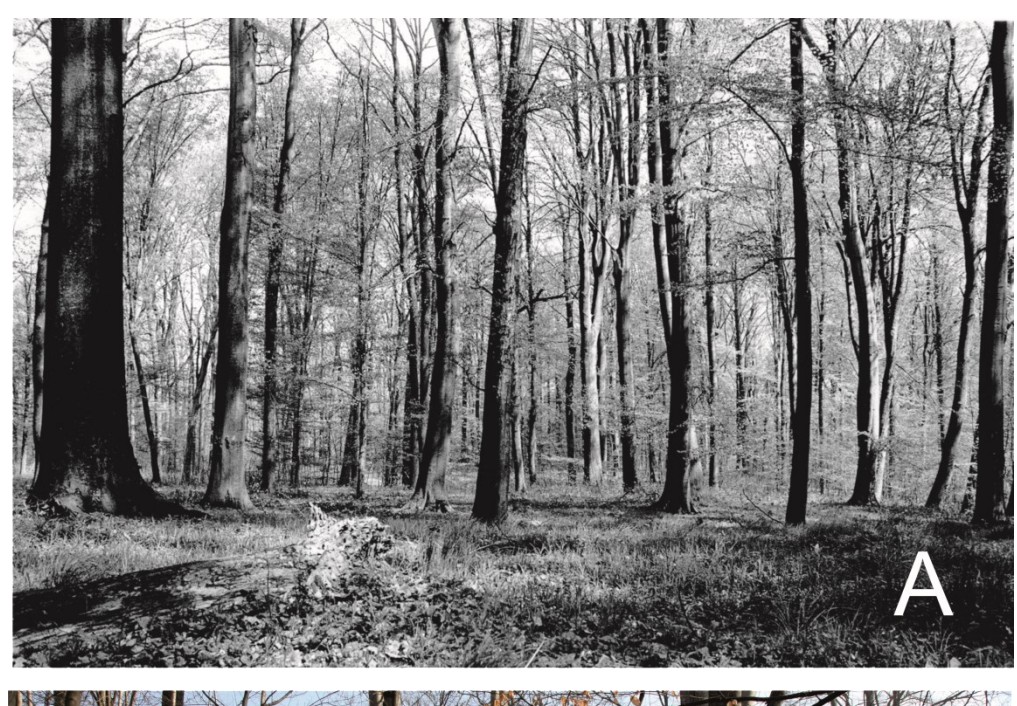

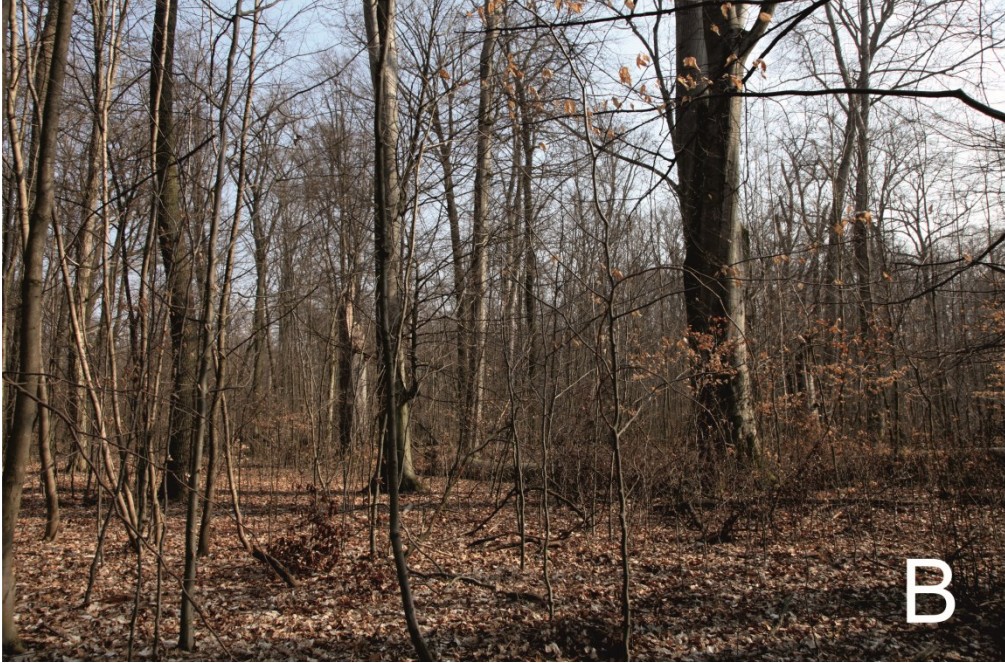

**Figure 5.** Changes in ground flora in plot J-III: (**A**)—In 1979, (**B**)—Present (2022).

## 3. Results

*3.1. Changes in the Species Composition and Abundance of Uropodina Communities in Jakubowo Nature Reserve in Subsequent Years*

The analysis of the communities in the subsequent years of the research project (1981–2022) revealed that out of the 17 Uropodina species found in total in the examined reserve, only 4 occurred regularly in the communities throughout the whole period: *O. minima*, *T. aegrota*, *T. pauperior* and *U. pannonica* (Table 1). The number of Uropodina species forming the communities in the consecutive periods of research varied from 6 to 12. The most species-rich community was found in 2006; it consisted of 12 species (i.e., 71%) of all identified species.

**Table 1.** Changes in species composition of Uropodina communities in Jakubowo nature reserve, 1981–2022.

| Species | 1981 | 1982 | 2005 | 2006 | 2012 | 2014 | 2016 | 2022 |
|---|---|---|---|---|---|---|---|---|
| *Olodiscus minima* (Kramer, 1882) | + | + | + | + | + | + | + | + |
| *Trachytes aegrota* (C. L. Koch, 1841) | + | + | + | + | + | + | + | + |
| *Trachytes pauperior* (Berlese, 1914) | + | + | + | + | + | + | + | + |
| *Urodiaspis pannonica* Willmann, 1952 | + | + | + | + | + | + | + | + |
| *Polyaspinus cylindricus* Berlese, 1916 | | | + | + | + | + | + | + |
| *Urodiaspis tecta* (Kramer, 1876) | + | + | + | + | + | | | + |
| *Oodinychus ovalis* (C. L. Koch, 1839) | | + | + | + | + | | | + |
| *Cilliba rafalskii* (Błoszyk Stachowiak et Halliday, 2008) | + | + | + | + | | | | |
| *Dinychus perforatus* Kramer, 1882 | | + | + | | | + | + | |
| *Dinychura cordieri* (Berlese, 1916) | | | + | + | | | | |
| *Phaulodiaspis rackei* (Oudemans, 1912) | | | | + | | + | | |
| *Discourella modesta* (Leonardi, 1889) | | | | | | | + | |
| *Uroobovella pyriformis* (Berlese, 1920) | | | | + | | | | |
| *Pulchellaobovella pulchella* (Berlese, 1904) | | | | + | | | | |
| *Neodiscopoma splendida* (Kramer, 1882) | | | | | | + | | |
| *Oodinychus obscurasimilis* (Hirschmann et Z.-Nicol, 1961) | | | | | | + | | |
| *Trachytes lamda* Berlese, 1903 | | + | | | | | | |
| Number of species | 6 | 9 | 10 | 12 | 7 | 9 | 7 | 7 |
| % of species | 35.29 | 52.94 | 58.82 | 70.59 | 41.18 | 52.94 | 41.18 | 41.18 |
| Number of samples | 120 | 120 | 288 | 360 | 120 | 120 | 120 | 120 |

Since 2005, *P. cylindricus* has been regularly found in the reserve, while *U. tecta* was not found in the period 2014–2016. Additionally, *O. ovalis* was only sporadically found in the reserve. The absence of these species in the analyzed samples does not indicate their complete disappearance, but it may also suggest a low abundance of the specimens, which made it impossible to catch them using the sampling methods used in this research. This is also relevant in the case of two other species, e.g., *C. rafalskii* and *D. perforatus*. The former was regularly found in the years 1981–2006, but after this period, it did not occur in the collected samples. The latter occurred in two periods, 1982–2005 and 2014–2016. Only in two research periods was the presence of *D. cordieri* and *Ph. rackei* recorded. The other species occurred extremely rarely in the reserve during the conducted research.

Figure 6 shows the observed changes in the abundance of four most numerous Uropodina species (i.e., *T. aegrota*, *T. pauperior*, *O. minima* and *U. pannonica*), which formed the community. Since 2012, a considerable decrease in the number of Uropodina has been observed in the analyzed samples. The differences in the average number of specimens per sample of these species have turned out to be statistically significant ((ANOVA) Kruskal–Wallis rank test H(7, N = 792) = 53.14, $p < 0.001$). They were observed in the following periods: 1981–2005 ***, 1981–2012 ***, 1981–2016 ***, 1982–2012 **, 2006–2012 **, 2012–2014 * and 2012–2022 ** (Table 2).

The average abundance of Uropodina in the reserve during the research periods remained at the level of 5400 specimens per m$^2$. In general, since 2006 in Jakubowo, one can observe a decrease in the abundance of Uropodina mites. Statistically significant differences in the average number of specimens per sample in the whole community of Uropodina in research periods 1981–2006 and 2021–2022 were observed ($p < 0.01$ Mann–Whitney U-test = 10.805, z = 2.88) (Table 3, Figure 7). Abundances below the average were observed in the years 1981, 2016, 2022, while abundances above the average were observed in the period 2005–2006. In the other periods, the abundance of Uropodina mites remained similar to the average (Figure 8).

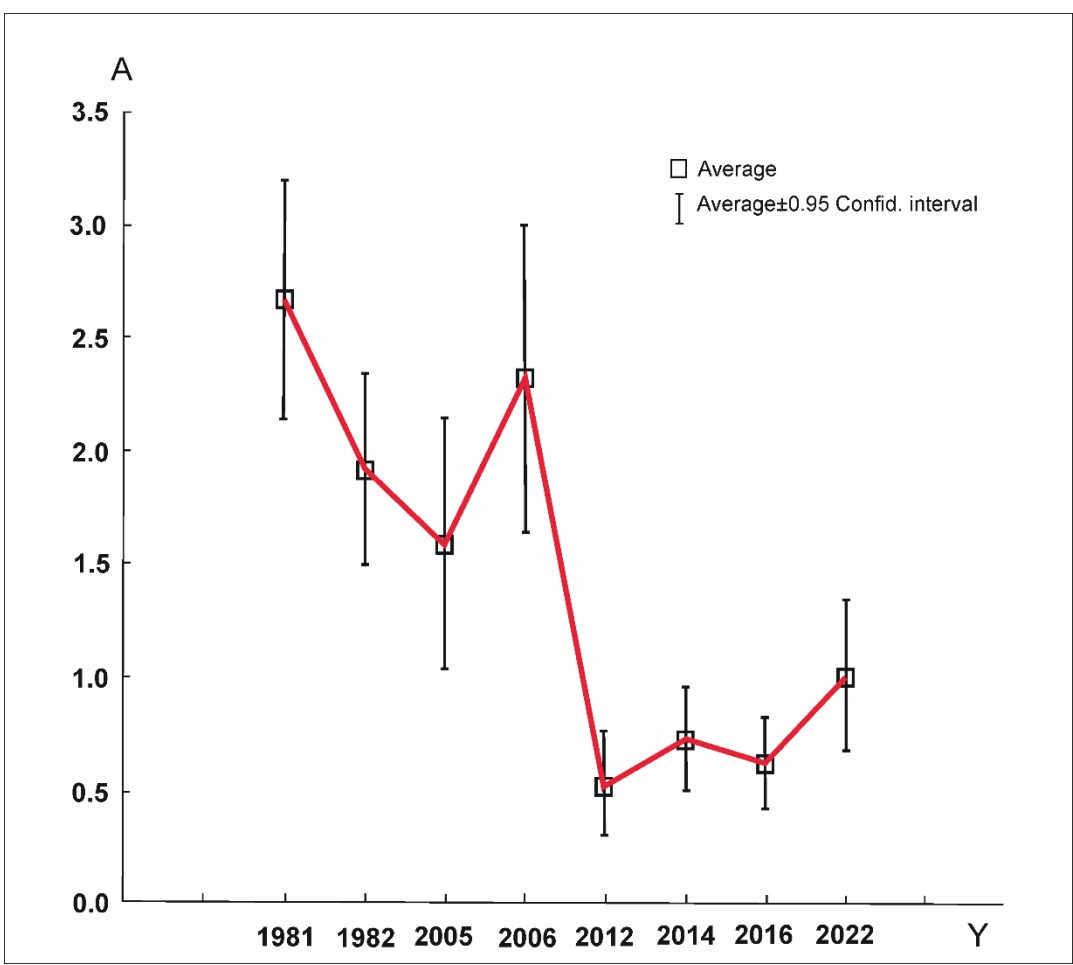

**Figure 6.** Dynamics of abundance (A) (specimen per sample) of four most numerous Uropodina species found in Jakubowo during consecutive research periods (Y).

**Table 2.** Statistical differences between the abundance of four most frequent species of Uropodina (i.e., *T. aegrota*, *T. pauperior*, *O. minima* and *U. pannonica*) during consecutive research periods. N—number of observations, X—average, SD—standard deviation, Med.—median.

|  |  | N | X | SD | Med. | X Rang |
|---|---|---|---|---|---|---|
| 1 | 1981 | 99 | 2.67 | 2.61 | 1 | 505.54 |
| 2 | 1982 | 99 | 1.92 | 2.13 | 1 | 425.34 |
| 3 | 2005 | 99 | 1.59 | 2.79 | 1 | 333.8 |
| 4 | 2006 | 99 | 2.31 | 3.37 | 1 | 421.41 |
| 5 | 2012 | 99 | 1.05 | 1.42 | 1 | 303.05 |
| 6 | 2014 | 99 | 1.45 | 1.24 | 1 | 405.2 |
| 7 | 2016 | 99 | 1.23 | 1.18 | 1 | 358.08 |
| 8 | 2022 | 99 | 1.73 | 1.68 | 1 | 419.58 |

**Table 3.** Statistical differences between the abundance of Uropodina in two research periods (1981–2006 and 2021–2022). N—number of observations, X—average, SD—standard deviation, Med.—median.

|  | N | X | SD | Med. | X Rang |
|---|---|---|---|---|---|
| 1981–2006 | 255 | 3.25 | 5.82 | 2 | 190.63 |
| 2012–2022 | 105 | 1.78 | 1.59 | 1 | 155.90 |

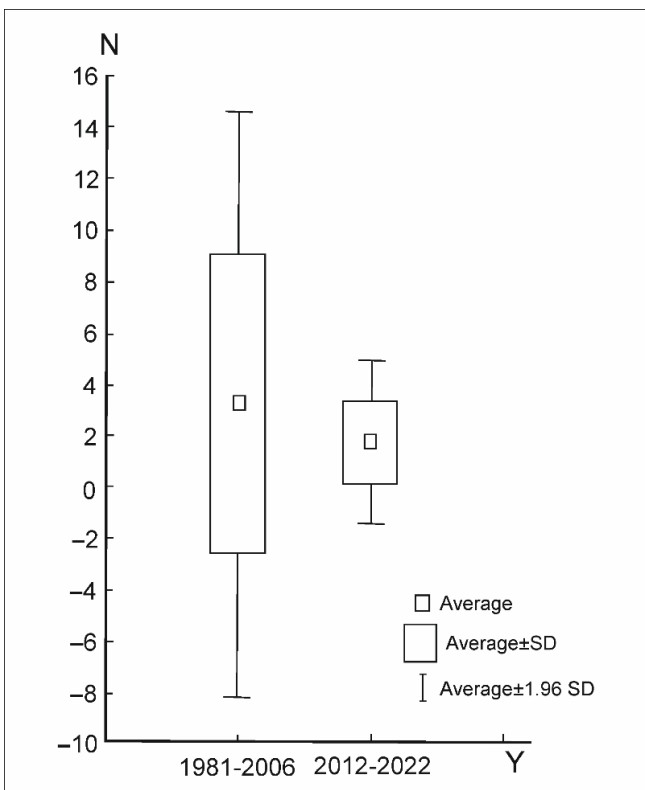

**Figure 7.** Average number (N) of Uropodina specimens in positive samples in the Jakubowo reserve in two research periods (Y) (1981–2006 and 2021–2022).

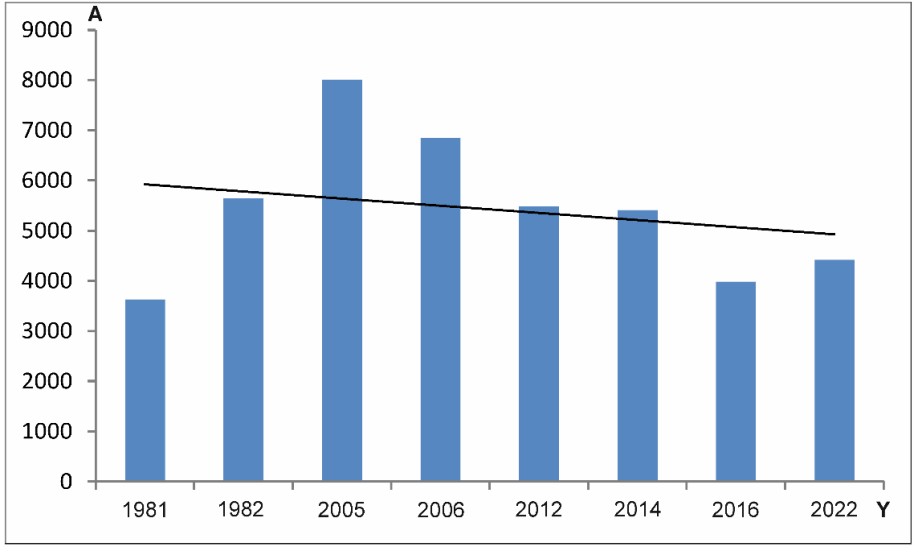

**Figure 8.** Changes in average abundance (A) (specimens per m$^2$) of Uropodina mites in Jakubowo over research years (Y).

The analysis also focuses on the changes in abundance of *T. aegrota*, *T. pauperior*, *O. minima* and *U. pannonica* (Figure 9), which were the four species with regular occurrence in the examined communities.

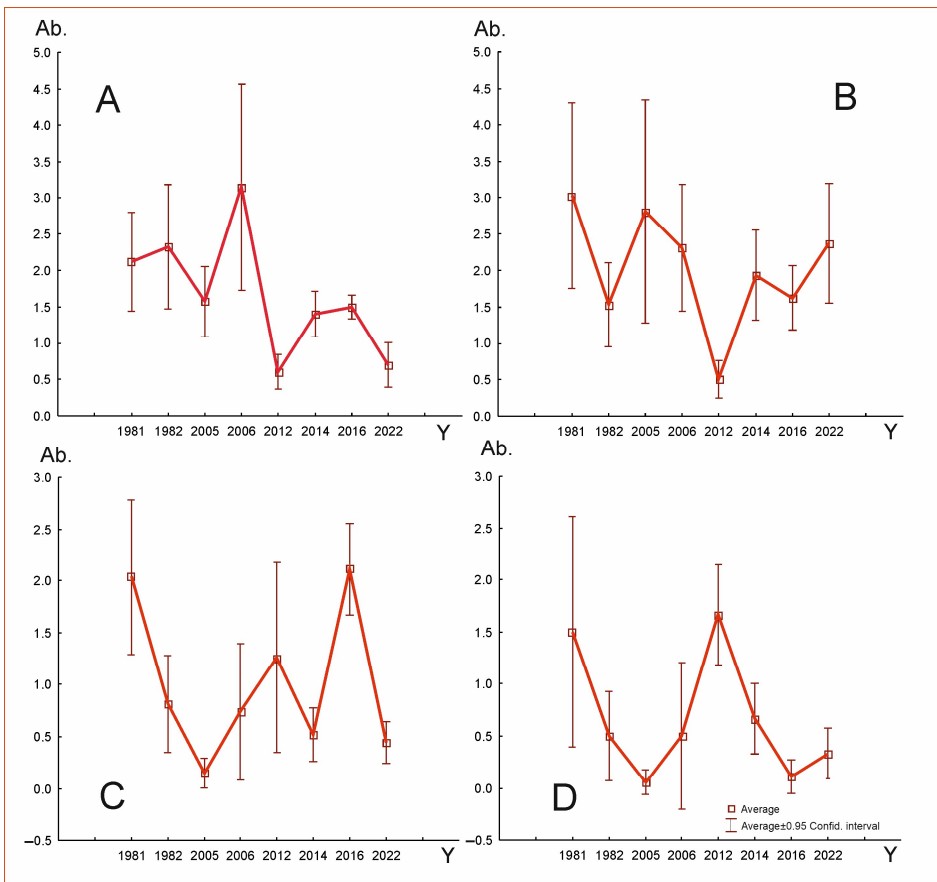

**Figure 9.** Changes in abundance (Ab.) over research years (Y) of four species, which regularly occurred in examined Uropodina communities in Jakubowo: (**A**)—*T. aegrota*, (**B**)—*O. minima*, (**C**)—*T. pauperior*, (**D**)—*U. pannonica*.

As shown in Figure 8, the curves indicate fluctuations in the abundance of the four regularly occurring Uropodina species in Jakubowo. The differences between the average values in the consecutive research periods for all species turned out to be statistically significant:

*O. minima*—(ANOVA) Kruskal–Wallis rank test H(7, N = 256) = 34.45; *p* < 0.00. In the case of this species, the average number of specimens was considerably different in the samples collected in the following periods: 1981–2012 ***, 2005–2012 **, 2006–2012 ***, 2012–2014 ***, 2012–2016 ** and 2012–2022 *** (Table 4).

**Table 4.** Statistical differences between the abundances of *O. minima* during consecutive research periods. N—number of observations, X—average, SD—standard deviation, Med.—median.

|  |  | N | X | SD | Med. | X Rang |
|---|---|---|---|---|---|---|
| 1 | 1981 | 32 | 3.03 | 3.55 | 1 | 149.59 |
| 2 | 1982 | 32 | 1.53 | 1.59 | 1 | 119 |
| 3 | 2005 | 32 | 2.81 | 4.27 | 1 | 131.02 |
| 4 | 2006 | 32 | 2.31 | 2.43 | 1 | 145.66 |
| 5 | 2012 | 32 | 0.5 | 0.72 | 0 | 61.25 |

**Table 4.** *Cont.*

|   |   | N | X | SD | Med. | X Rang |
|---|---|---|---|---|---|---|
| 6 | 2014 | 32 | 1.94 | 1.7 | 1 | 140.88 |
| 7 | 2016 | 32 | 1.63 | 1.24 | 1 | 132.81 |
| 8 | 2022 | 32 | 2.38 | 2.27 | 1 | 147.8 |

*T. aegrota*—(ANOVA) Kruskal–Wallis rank test H(7, N = 320) = 49.33; *p* < 0.001. A significant difference in the average number of specimens was observed in the following periods: 1981–2012 **, 1981–2022 **, 1982–2012 ***, 1982–2022 **, 2005–2012 *, 2005–2022 *, 2006–2012 ***, 2006–2022 ***, 2012–2014 *, 2012–2016 ***, 2014–2022 * and 2016–2022 *** (Table 5).

**Table 5.** Statistical differences between the abundances of *T. aegrota* during consecutive research periods. N—number of observations, X—average, SD—standard deviation, Med.—Median.

|   |   | N | X | SD | Med. | X Rang |
|---|---|---|---|---|---|---|
| 1 | 1981 | 40 | 2.13 | 2.13 | 1 | 182.85 |
| 2 | 1982 | 40 | 2.33 | 2.66 | 1 | 183.96 |
| 3 | 2005 | 40 | 1.58 | 1.52 | 1 | 165.92 |
| 4 | 2006 | 40 | 3.15 | 4.44 | 1 | 195.74 |
| 5 | 2012 | 40 | 0.6 | 0.74 | 0 | 98.05 |
| 6 | 2014 | 40 | 1.4 | 0.98 | 1 | 169.25 |
| 7 | 2016 | 40 | 1.5 | 0.51 | 1.5 | 187.25 |
| 8 | 2022 | 40 | 0.7 | 0.97 | 0.5 | 100.98 |

*T. pauperior*—(ANOVA) Kruskal–Wallis rank test H(7, N = 216) = 57.81; *p* < 0.001. The average number of specimens per sample was considerably different in the following periods: 1981–2005 ***, 1981–2006 **, 1981–2014 *, 1981–2022 *, 1982–2016 **, 2005–2016 ***, 2006–2016 ***, 2012–2016 ***, 2014–2016 *** and 2016–2022 *** (Table 6).

**Table 6.** Statistical differences between the abundances of *T. pauperior* during consecutive research periods. N—number of observations, X—average, SD—standard deviation, Med.—median.

|   |   | N | X | SD | Med. | X Rang |
|---|---|---|---|---|---|---|
| 1 | 1981 | 27 | 2.04 | 1.89 | 1 | 151.94 |
| 2 | 1982 | 27 | 0.81 | 1.18 | 0 | 104.07 |
| 3 | 2005 | 27 | 0.15 | 0.36 | 0 | 66.24 |
| 4 | 2006 | 27 | 0.74 | 1.65 | 0 | 90.11 |
| 5 | 2012 | 27 | 1.26 | 2.31 | 0 | 99.2 |
| 6 | 2014 | 27 | 0.52 | 0.64 | 0 | 94.76 |
| 7 | 2016 | 27 | 2.11 | 1.12 | 2 | 169.94 |
| 8 | 2022 | 27 | 0.44 | 0.51 | 0 | 91.72 |

*U. pannonica*—(ANOVA) Kruskal–Wallis rank test H97, N = 144) = 40.72; *p* < 0.001. The average number of specimens per sample was considerably different in the following periods: 1981–2005 *, 1982–2012 **, 2005–2012 ***, 2006–2012 ***, 2012–2016 *** and 2012–2022 ** (Table 7).

**Table 7.** Statistical differences between the abundance of *U. pannonica* during consecutive research periods. N—number of observations, X—average, SD—standard deviation, Med.—median.

|   |      | N  | X    | SD   | Med. | X Rang |
|---|------|----|------|------|------|--------|
| 1 | 1981 | 18 | 1.5  | 2.23 | 1    | 90.89  |
| 2 | 1982 | 18 | 0.5  | 0.86 | 0    | 67.64  |
| 3 | 2005 | 18 | 0.06 | 0.24 | 0    | 47     |
| 4 | 2006 | 18 | 0.5  | 1.42 | 0    | 59.53  |
| 5 | 2012 | 18 | 1.67 | 0.97 | 1    | 118.67 |
| 6 | 2014 | 18 | 0.67 | 0.69 | 1    | 81.28  |
| 7 | 2016 | 18 | 0.11 | 0.32 | 0    | 50.5   |
| 8 | 2022 | 18 | 0.33 | 0.49 | 0    | 64.5   |

*3.2. Dynamics of Species Composition and Abundance in Uropodina Communities in Examined Ground Plots in Jakubowo Nature Reserve*

Different biotic and abiotic conditions prevailing in the examined ground plots resulted in the formation of Uropodina communities with different species composition and structure (Table 8). During the research project, out of 17 Uropodina species found in the reserve, only 7 of them were present in all three monitored plots. They were the following species: *O. minima*, *T. aegrota*, *T. pauperior*, *U. pannonica*, *U. tecta*, *C. rafalskii* and *O. ovalis*. The species found only in plot J-I were *T. lamda* and *O. obsurasimilis*. On the other hand, in plot J-III, there were such species as follows: *N. splendida*, *D. modesta*, *U. pyriformis* and *D. cordieri*. No exclusive species were found in plot J-II.

The highest index of species similarity (S = 62%) was recorded for plots J-II and J-III, whereas the most divergent species composition (S = 44%) was revealed in plots J-I and J-II (see Figure 10).

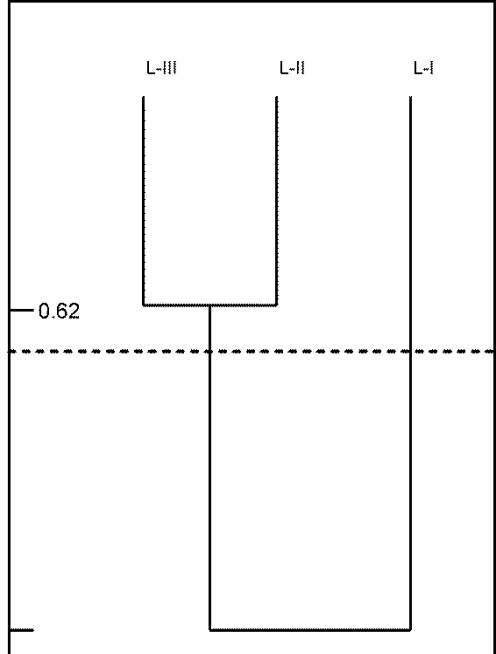

**Figure 10.** Species similarity (S) in examined ground plots (I, II, III) in Jakubowo nature reserve.

**Table 8.** Structure of dominance (D%) and frequency of occurrence (F%) of the found Uropodina species in monitored ground plots (J-I, J-II, J-III) in Jakubowo nature reserve during research periods. Bold – dominant and the most frequent species.

| Species | 1981 D% | 1981 F% | 1982 D% | 1982 F% | 2005 D% | 2005 F% | 2006 D% | 2006 F% | 2012 D% | 2012 F% | 2014 D% | 2014 F% | 2016 D% | 2016 F% | 2022 D% | 2022 F% |
|---|---|---|---|---|---|---|---|---|---|---|---|---|---|---|---|---|
| **J-I** | | | | | | | | | | | | | | | | |
| *O. minima* | **57.89** | 25.0 | **39.34** | 22.50 | **54.06** | **57.29** | **46.74** | 38.33 | 17.14 | 12.50 | **54.76** | 33.33 | 26.67 | **60.00** | **63.08** | 46.67 |
| *T. aegrota* | 22.81 | 20.0 | **45.90** | 32.50 | **36.11** | 46.88 | 29.35 | 30.83 | **31.43** | 25.00 | 23.81 | 20.00 | 23.33 | **50.00** | 3.08 | 6.67 |
| *U. pannonica* | 10.53 | 7.5 | 1.64 | 2.50 | 4.73 | 8.33 | 8.42 | 9.17 | **42.86** | 28.13 | 7.14 | 6.67 | 3.33 | 10.00 | 4.62 | 10.00 |
| *T. pauperior* | 7.02 | 7.5 | | | 1.13 | 4.17 | 4.89 | 6.67 | 2.86 | 3.13 | 2.38 | 3.33 | **43.33** | **50.00** | 4.62 | 10.00 |
| *C. rafalskii* | 1.75 | 2.5 | 1.64 | 2.50 | | | | | | | | | | | | |
| *U. tecta* | | | 4.92 | 5.00 | 2.08 | 7.29 | 7.61 | 10.83 | | | | | | | 6.15 | 10.00 |
| *O. ovalis* | | | 4.92 | 2.50 | 0.57 | 1.04 | 0.27 | 0.83 | | | | | | | 4.62 | 6.67 |
| *T. lamda* | | | 1.64 | 2.50 | | | | | | | | | | | | |
| *P. cylindricus* | | | | | 1.13 | 6.25 | 2.17 | 5.00 | 5.71 | 6.25 | 7.14 | 6.67 | 3.33 | 10.00 | 13.85 | 23.33 |
| *D. cordieri* | | | | | 0.19 | 1.04 | | | | | | | | | | |
| *P. pulchella* | | | | | | | 0.27 | 0.83 | | | | | | | | |
| *Ph. rackei* | | | | | | | 0.27 | 0.83 | | | 2.38 | 3.33 | | | | |
| *O. obscurasimilis* | | | | | | | | | | | 2.38 | 3.33 | | | | |
| Number of species | 5 | | 7 | | 8 | | 9 | | 5 | | 7 | | 5 | | 7 | |
| **J-II** | | | | | | | | | | | | | | | | |
| *T. pauperior* | **46.51** | 17.50 | 16.67 | 5.00 | 4.66 | 6.25 | 5.59 | 4.17 | | | | | 25.00 | 10.00 | 11.11 | 4.76 |
| *T. aegrota* | 23.26 | **17.50** | **33.33** | 7.50 | 23.83 | 26.04 | 16.78 | 15.83 | | | | | | | | |
| *O. minima* | 18.60 | 17.50 | **33.33** | 10.00 | **54.40** | 38.54 | **50.35** | 31.67 | | No data | | | **75.00** | 30.00 | **88.89** | 28.57 |
| *U. tecta* | 4.65 | 5.00 | | | 12.95 | 9.38 | 17.48 | 10.83 | | | | | | | | |
| *U. pannonica* | 4.65 | 5.00 | | | 1.04 | 2.08 | 2.10 | 1.67 | | | | | | | | |
| *C. rafalskii* | 2.33 | 2.50 | 8.33 | 2.50 | 0.52 | 1.04 | 3.50 | 3.33 | | | | | | | | |
| *D. perforatus* | | | 8.33 | 2.50 | 0.52 | 1.04 | | | | | | | | | | |
| *O. ovalis* | | | | | 2.07 | 3.13 | 4.20 | 5.00 | | | | | | | | |
| Number of species | 6 | | 5 | | 8 | | 7 | | | | | | 2 | | 2 | |
| **J-III** | | | | | | | | | | | | | | | | |
| *T. aegrota* | 35.63 | 45.00 | **54.95** | 45.00 | **64.78** | 42.71 | **63.71** | **51.67** | 4.00 | 3.33 | **48.65** | 33.33 | 23.53 | **50.00** | 24.00 | 38.10 |
| *O. minima* | 32.18 | 37.50 | 18.92 | 32.50 | 14.95 | 26.04 | 12.79 | 24.17 | 8.00 | 6.67 | 21.62 | 20.00 | **44.12** | **60.00** | **54.00** | 52.38 |
| *T. pauperior* | 17.82 | 32.50 | 18.02 | 27.50 | 3.65 | 9.38 | 2.61 | 7.50 | **64.00** | 13.33 | 16.22 | 16.67 | 14.71 | 30.00 | 4.00 | 9.52 |
| *U. pannonica* | 10.92 | 15.00 | 8.00 | 7.21 | 1.99 | 5.21 | 1.83 | 5.00 | | | 8.11 | 10.00 | | | | |
| *U. tecta* | 2.87 | 10.00 | 0.90 | 2.50 | 5.98 | 9.38 | 12.01 | 21.67 | 8.00 | 6.67 | | | | | 4.00 | 4.76 |
| *C. rafalskii* | 0.57 | 2.50 | | | | | | | | | | | | | | |
| *P. cylindricus* | | | | | | | | | 4.00 | 3.33 | | | 11.76 | 20.00 | 12.00 | 23.81 |
| *O. ovalis* | | | | | 8.64 | 13.54 | 6.53 | 13.33 | 12.00 | 3.33 | | | | | 2.00 | 4.76 |
| *D. perforatus* | | | | | | | | | | | 2.70 | 3.33 | 2.94 | 10.00 | | |
| *N. splendida* | | | | | | | | | | | 2.70 | 3.33 | | | | |
| *D. modesta* | | | | | | | | | | | | | 2.94 | 10.00 | | |
| *U. pyriformis* | | | | | | | 0.26 | 0.83 | | | | | | | | |
| *D. cordieri* | | | | | | | 0.26 | 0.83 | | | | | | | | |
| Number of species | 6 | | 5 | | 6 | | 7 | | 6 | | 6 | | 6 | | 6 | |

Figure 11 shows the changes in the abundances of the three most numerous Uropodina species in the reserve during the research periods in plots J-I and J-III. In the case of *O.*

*minima* and *T. pauperior*, one can observe an increase in the abundance, while in the case of *T. aegrot*, the long-term diagram shows a decrease.

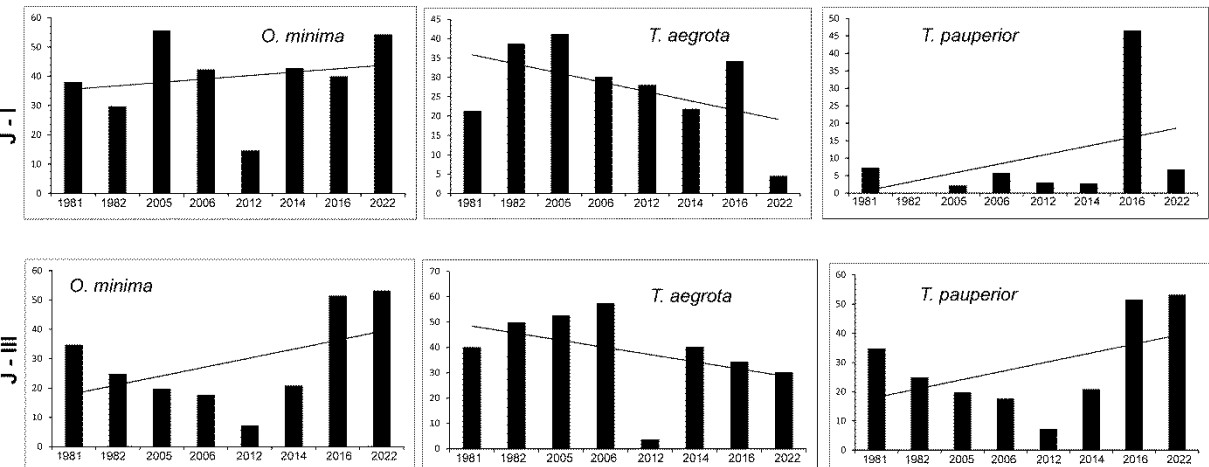

**Figure 11.** Dynamics of changes in abundances of three most abundant species (average number of specimens per sample) in consecutive research periods (horizontal lines indicate trends) in examined ground plots (J-I, J-III) in Jakubowo nature reserve.

## 4. Discussion

One of the characteristics of local Uropodina communities in such small forest complexes is the low number of species [3,4]. Out of the 33 species found so far in the oak-hornbeam forests in Wielkopolska, [3], as many as 17 species (51%) were found in the examined location, with the maximum number of species found only in one year, which was 12 species (36%). Thus, one can conclude that the area of Jakubowo nature reserve, despite its small size (4.22 ha), offers relatively favorable living conditions for soil mite communities from this group. The examined communities inhabiting the reserve were dominated by parthenogenetic and eurytopic soil species: *O. minima*, *T. aegrota*, *T. pauperior* and *U. pannonica* [1,15]. These species can be considered a permanent element of the whole community, as they were present throughout all research periods. The occurrence of other species was associated with specific conditions concerning humidity and temperature, resulting from the diversity of the plant cover or, in some cases, the presence of specific micro-environments inhabited by these species. One of these species is associated with the mole's nests, which occasionally appeared in the examined ground plots, i.e., *Ph rackei* [1,8,16], which was found twice during the research. Moreover, in 2006, the presence of *U. pyriformis* and *P. pulchella*, which inhabit dead wood and hollows, was also recorded [1,8,9], probably due to the presence of an increasing amount of dead wood in the areas under scrutiny.

The natural succession of plant cover observed in each ground plot, especially in plot J-I and J-II, undoubtedly has a bearing on the changes in the communities of the discussed group of mites, both in the species diversity and abundance [4]. The observed trend of an increase in the number of *O. minima* and *T. pauperior*, i.e., species with higher moisture requirements [1], in the reserve is most likely caused by the formation of an undergrowth layer, which causes additional shading and helps maintain higher soil moisture. These species compete with *T. aegrota*, which causes a decreasing trend in the abundance of this species. It is noteworthy that the percentage of *O. ovalis*, which is currently one of the most numerous species of Uropodina in Poland [17], is apparently low here. Furthermore, the global warming effect may have been responsible for the appearance in the reserve in 2014 of two species not previously listed here, i.e., *O. obscurasimilis* and *N. splendida*. Perhaps their presence is the result of broadening of the ranges of these species. The former is a species, which occurs in the Carpathian Mountains, and its range extends to the north

along the Vistula line, whereas the latter has a south-European and boreal-mountain range of occurrence [1]. However, this phenomenon requires further research.

To sum up, one can conclude that the conducted research confirmed the previous observations on the functioning of the Uropodina communities in Jakubowo nature reserve over the course of 30 years [4]. The obtained results corroborate both the permanent loss of stenotopic and rare species, such as *C. rafalskii* and *T. lamda*, which were not recorded even once after 1982 [4]. Both species are rare and sparse: *Trachytes lamda* Berlese, 1903, has the CR category, according to the Red List for Uropodina of Poland, while *C. rafalski* has the EN category [18]. *Trachytes lamda* was recorded in the study area in the late 1970s but was no longer recorded in the 1980s [1]. It is a species typical for oak-hornbeam forests and this area, as it was still numerous in the 1980s in the nearby area; however, it has not been recorded there since 2005 [1,4]. Moreover, the previously observed decline in the total number of Uropodina in the litter and soil in the examined areas [4] can now be considered a well-established and unfavorable trend, which has been observed for almost 10 years now. These unfavorable changes are undoubtedly the result of the progressive negative anthropogenic processes observed in the examined location.

## 5. Conclusions

The results obtained from the conducted research project clearly show the importance of constant monitoring of changes taking place in the soil environment of forests. Only in this way is it possible to determine the impact of various factors, both global (climate changes) and local (environmental contamination, plant succession changes, etc.), on the soil environment and the soil fauna that inhabits them to plan any necessary protective measures.

**Author Contributions:** Conceptualization, J.B. and A.N.; Methodology, A.N. and J.B.; Software, J.B.; Validation, J.B., A.N., M.Z. and M.K.; Formal Analysis, J.B., A.N., M.Z. and M.K.; Investigation, J.B., A.N., M.K. and M.Z.; Resources, J.B., A.N.and M.Z.; Data Curation, J.B.; Writing—Original Draft Preparation, J.B. and A.N.; Writing—Review and Editing, A.N.; Visualization, J.B. and A.N.; Supervision, A.N.; Project Administration, J.B.; Funding Acquisition, J.B. and A.N. All authors have read and agreed to the published version of the manuscript.

**Funding:** This work was possible due to financial support from the Department of General Zoology AMU.

**Institutional Review Board Statement:** Not applicable.

**Informed Consent Statement:** Not applicable.

**Data Availability Statement:** The data presented in this study are stored in a computer database called AMUNATCOLL and openly available at: https://amunatcoll.pl/ (accessed on 25 April 2022).

**Acknowledgments:** The research was conducted within a project: "*Inicjatywa Doskonałosci—Uczelnia badawcza*" (No. 049/34/D-UB/0063). The authors would like to thank all staff members and students who contributed to the collection of the material for 43 years of the research project. Special words of gratitude are owed to Ziemowit Olszanowski, Marek Kaliszewski, Tomasz Rutkowski, Jacek Wendzonka Ireneusz Chojnacki, Jarosław Fiszer, Bogdan Rzeszuto and Edward Chwieduk. Moreover, we would like to thank Ireneusz Chojnacki for producing photographic documentation in 1979.

**Conflicts of Interest:** The authors declare no conflict of interest.

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
