# Peer review of "Changes in Forest Stand and Stability of Uropodine Mites Communities (Acari: Parasitiformes) in Jakubowo Nature Reserve in the Light of Long-Term Research"

_forests, doi:10.3390/f13081219_

Round 1

Reviewer 1 Report

Major revision, because:

Line 278

Did the authors calculate the stock of deadwood in 1982 and in 2004-2012?

Line 287

This moment is doubtful, because the authors do not provide climatic data when describing the model sites.

Methodology:

Why did the authors not conduct research during the period 1990-2003?

The article discusses the period from 1981 to 2022. It is more logical to present data for each year, either to be limited to five-year periods, or ten-year ones. The presented method weakly proves the change in the fauna of the studied group of spiders in the indicated periods. For such studies, it is necessary to analyze the annual and interannual dynamics, using climate data and other abiotic factors.

Author Response

Changes in forest stand and stability of Uropodina communities (Acari: Parasitiformes) in Jakubowo nature reserve in the light of long-term research

The authors of the study are grateful to the Reviewer for all comments and suggestions. All of them have turned out to be extremely helpful, which obviously has considerably improved the overall quality of the manuscript.

Detailed responses to the Reviewer comments: 

Line 278

Did the authors calculate the stock of deadwood in 1982 and in 2004-2012?

- Until 1983, fallen trees were removed from the reserve. For this reason, we did not estimate the exact amount of dead wood in the examined plots. The aim of this study was to show that leaving dead wood can have impact on the emergence of new species in a community.

Line 287

This moment is doubtful, because the authors do not provide climatic data when describing the model sites.

- In the sentence, we emphasize that global climate change may have an impact on the expansion of the ranges of two southern and mountain species: O. obscurasimilis and N. splendida. However, confirmation of this phenomenon requires more detailed research, which we also write about later.

We have climatic data for the studied plot, but they will be used in subsequent studies published in the future.

Methodology:

Why did the authors not conduct research during the period 1990-2003?

The article discusses the period from 1981 to 2022. It is more logical to present data for each year, either to be limited to five-year periods, or ten-year ones. The presented method weakly proves the change in the fauna of the studied group of spiders in the indicated periods. For such studies, it is necessary to analyze the annual and interannual dynamics, using climate data and other abiotic factors.

  • At the beginning of the research in the 1980s, it was not planned that the observations would be conducted for so long (over 40 years!). In the initial period of the study, the samples were collected every two weeks each year, in the years 2005-2006, or in 2012 and 2014, they were collected regularly every month. In the following years, the research was conducted depending on the projects carried out and available funds. Only a small part of the material was used in the publication (the rest is currently under development), but the samples were selected so that in the following years they would overlap with the dates. This makes it possible to trace the dynamics of the examined communities over a longer period of time against the background of changing vegetation cover and other factors. In this work, the aim was to show the phenomenon initially, while a detailed study, such as the published research from the period 1979-1983 (BÅ‚oszyk 1999), will be published after the end of the current monthly research sessions.

Reviewer 2 Report

Long-term studies of communities are very important and rare. There is no doubt that the data obtained by the authors during 40-year monitoring of mite fauna of the same nature reserve should be published. But it is doubtful that the journal Forests is suitable for this publication. The data themselves seem to be more valuable than the conclusions. Probably, it would be more appropriate to publish this article in Data journal.

Comments:

Line 2-3 I would recommend to include the word "mite" or "mites" to the title. It would help the readers which are not expert in this field to understand what is the article about.

Line 17-18  The conclusion "The most important phenomenon observed during the research period was a considerable decrease in the abundance of Uropodina" is not supported with the data presented in the article. According to Figure 5 the dynamics of abundance (specimen per sample) could reflect just fluctuations. Moreover, the abundance is growing in 2012-2022. And there is no reliable difference in abundance in 2005 and 2022.

Line 41 If the current article is a continuation of the previous research (Line 41), please point out what is novelty of this particular article.

Line 19. The study was started in 1981. If  the species was recorded in 1981, but then was not recorded in 1982-2022, it does not mean that the species is "rare" or "stenotopic". It could just indicate that the species is not typical for the territory, and its finding in 1981 was accidental.

Lines 20-21. It is not quite clear what period do you mean: 36 years before 1982 or after it.

Line 58 It is doubtful that it is possible to count or even to assess "the number of Uropodina communities". One researcher would find many communities, while another - just one, because different researchers have different answers on the question what is "community".

Line 64 and 69 Could you please to describe "biocenometer" and "Tullgren funnels" in brief and to provide a references about these devises.

Line 80-85. Please explain clearer what percentage of dominance do you mean. Do you mean percentage of samples in which the species is dominant? The same question about frequency.

Figure 1 Please, provide the geographic coordinates of each plot. The figure could be removed, since it is not informative. It is enough to indicate that the plots are in the center of the nature reserve.

Line 153 Please delete "Among these there were species like" and replace it with a column. ... 4 occurred regularly in the communities throughout the whole period: O. minima, T. aegrota, T. pauperior and U. pannonica.

Line 156 "The most species-rich community was found 155 during in 2006, it consisted of 12 species. " - This is just because the highest number of samples: 360 samples were collected in 2006, while only 120 samples in each subsequent year.

Line 186 "In general, since 2006 in Jakubowo, one can 186 observe a decrease in the abundance of Uropodina mites." - This conclusion is not supported with the data presented in the article. As is shown in Figure 5, there is no reliable difference in abundance in 2005 and 2022.

Figure 7 also does not support the conclusion about the decrease of abundance. Only fluctuation of abundance of four dominant species are seen.

Line 265-267 "...the area of Jakubowo nature reserve, despite its small area (4.22 ha), offers relatively favourable living conditions for soil mite communities..." There is a formula: the dependence of number of species from the area of the region. This dependence is not linear. Therefore, the presented comparison of species richness of larger territories do not prove this conclusion. The conclusion about favourable living conditions could be proved by comparison with other territories of the same area.

Author Response

Changes in forest stand and stability of Uropodina communities (Acari: Parasitiformes) in Jakubowo nature reserve in the light of long-term research

The authors of the study are grateful to the Reviewer for all comments and suggestions. All of them have turned out to be extremely helpful, which obviously has considerably improved the overall quality of the manuscript.

Detailed responses to the Reviewer comments: 

Line 2-3 I would recommend to include the word "mite" or "mites" to the title. It would help the readers which are not expert in this field to understand what is the article about.

  • The suggested change in the title has been taken into account and all necessary corrections have been made.

Line 17-18  The conclusion "The most important phenomenon observed during the research period was a considerable decrease in the abundance of Uropodina" is not supported with the data presented in the article. According to Figure 5 the dynamics of abundance (specimen per sample) could reflect just fluctuations. Moreover, the abundance is growing in 2012-2022. And there is no reliable difference in abundance in 2005 and 2022.

- The results also tested the average number of the whole community of Uropodina in Jakubowo reserve in two research periods (1981-2006 and 2021-2022). The test results turned out to be statistically significant (p <0.01 Mann-Whitney U rank test U = 10805, z = 2.88), which confirms that the size of the entire group decreased in the later period of the study (Figure 6, Table 3).

Line 19. The study was started in 1981. If  the species was recorded in 1981, but then was not recorded in 1982-2022, it does not mean that the species is "rare" or "stenotopic". It could just indicate that the species is not typical for the territory, and its finding in 1981 was accidental.

  • The species T. lamda was recorded in this area (Jakubowo reserve) in the 1970s (BÅ‚oszyk 1999), but it was not found during our research. It is a species typical for oak-hornbeam forests, as it was relatively numerous in the nearby, also oak-hornbeam forest reserve, Las GrÄ…dowy at Mogilnica (BÅ‚oszyk 1999, NapieraÅ‚a et al. 2014). In this location the species was numerous until 1983. Also there it has not been found since 2005 (NapieraÅ‚a et al. 2014). The second rare species that does not occur in Jakubowo reserve after 2006 is Cilliba rafalski. This species previously occurred regularly in the study site, in one plot, but after changes in the plant cover at this site, it was not found to be present. On a national scale, it is a rare and stenotopic species (NapieraÅ‚a et al. 2018).

Lines 20-21. It is not quite clear what period do you mean: 36 years before 1982 or after it.

  • All necessary information has been provided in the manuscript.

Line 41 If the current article is a continuation of the previous research (Line 41), please point out what is novelty of this particular article.

  • All necessary information has been provided in the manuscript.

Line 58 It is doubtful that it is possible to count or even to assess "the number of Uropodina communities". One researcher would find many communities, while another - just one, because different researchers have different answers on the question what is "community".

  • The authors meant “abundance of mites in communities”. The sentence has been corrected.

Line 64 and 69 Could you please to describe "biocenometer" and "Tullgren funnels" in brief and to provide a references about these devises.

  • Tullgren funnels, also known as Berlese trap, or Berlese-Tullgren funnel, is an apparatus used to extract living organisms, particularly arthropods, from soil samples. This type of extraction is commonly used in acarology and was described by Antonio Berlese in 1905. (Berlese, Antonio (1905). Apparecchio per raccogliere presto ed in gran numero piccoli Artropodi [Apparatus for gathering early and in large numbers small arthropods] (in Italian). Pp. 85-90.). A biocenometer (frame or cylinder) is a tool for taking soil samples with a specific area. Both tools are standardly used in soil fauna research and are widely known, therefore, there is no need to describe them in acarological publications.

Line 80-85. Please explain clearer what percentage of dominance do you mean. Do you mean percentage of samples in which the species is dominant? The same question about frequency.

  • Dominance and frequency are commonly used indexes in acarological (and other ecological) studies (see e.g. BÅ‚oszyk 1999; Magurran 2004). Dominance is the number of specimens of particular species/number of all specimens of the community. Frequency is the number of samples where the species was found/number of all collected samples.

Figure 1 Please, provide the geographic coordinates of each plot. The figure could be removed, since it is not informative. It is enough to indicate that the plots are in the center of the nature reserve.

  • The geographic coordinates of each plot have been provided. Figure 1 has been corrected.

Line 153 Please delete "Among these there were species like" and replace it with a column. ... 4 occurred regularly in the communities throughout the whole period: O. minima, T. aegrota, T. pauperior and U. pannonica.

  • The sentence has been corrected as suggested by the reviewer.

Line 156 "The most species-rich community was found 155 during in 2006, it consisted of 12 species. " - This is just because the highest number of samples: 360 samples were collected in 2006, while only 120 samples in each subsequent year.

- To some extent, a higher number of samples have an impact on the greater number of species found. However, as can be seen, the threefold increase in the number of trials slightly increased the number of species found. In 1982 and 2014, when 120 samples were collected, the number of species was 9, which is only 3 less than in 2006 (with 360 samples). This is due to the fact that, as shown in previous studies (Błoszyk 1999), already the number of 30 samples is sufficient to catch most of the species in a community. With additional samples, the number of species increases slightly as rare species are also collected. This is also confirmed by studies comparing the effectiveness of quantitative and qualitative methods in determining the number of species (Błoszyk J., Napierała A. 2018. Community structure of mesofauna in the light of qualitative and quantitative research on soil mites. European Journal of Biological Research, 8 (4) : 252-262. Http://dx.doi.org/10.5281/zenodo.2248744). In this case, the material was simultaneously collected with qualitative methods (litter screening), which allowed to state that the quantitative material included all species found in the reserve in a given year on the studied plot. Detailed analyses will be presented in the forthcoming publications.

Line 186 "In general, since 2006 in Jakubowo, one can  observe a decrease in the abundance of Uropodina mites." - This conclusion is not supported with the data presented in the article. As is shown in Figure 5, there is no reliable difference in abundance in 2005 and 2022.

  • The results of the test of average number of the whole community of Uropodina in Jakubowo reserve in two research periods (1981-2006 and 2021-2022) proved to be statistically significant (p <0.01 U Mann-Whitney's U test U = 10805, z = 2.88) (Figure 6, Table 3).

Figure 7 also does not support the conclusion about the decrease of abundance. Only fluctuation of abundance of four dominant species are seen.

  • In the case of the four dominant species, fluctuations in abundance were indeed observed, which it clearly stated in the text, while the size of the entire community decreased (Fig. 6 and the test of differences in abundance in the 1981-2006 and 2012-2022 periods have been added.

Line 265-267 "...the area of Jakubowo nature reserve, despite its small area (4.22 ha), offers relatively favourable living conditions for soil mite communities..." There is a formula: the dependence of number of species from the area of the region. This dependence is not linear. Therefore, the presented comparison of species richness of larger territories do not prove this conclusion. The conclusion about favourable living conditions could be proved by comparison with other territories of the same area.

  • In this comparison, we did not mean the area itself, but the diversity of habitats, age and type of tree stand, which is a convenient habitat for the examined mites. For comparison - in litter and soil of the legally protected oak-hornbeam forests (nature reserves and the Wielkopolski National Park), 26 species of Uropodina were recorded, (NapieraÅ‚a 2008 I NapieraÅ‚a et al. 2009). Therefore, it turns out that in such a small reserve - three areas to be precise (3x650m), as many as 17 species were found.

Round 2

Reviewer 1 Report

Accept present form. I have no comments. The authors accepted all corrections and responded to my comments.

Author Response

Changes in Forest Stand and Stability of Uropodine Mites Communities (Acari: Parasitiformes) in Jakubowo Nature Reserve in the Light of Long-term Research

The authors of the study are grateful to the Reviewer for comments and suggestions. All of them have turned out to be extremely helpful, which obviously has considerably improved the overall quality of the manuscript. New figures of better quality have been added in the manuscript.